# Differences in the Course of Depression and Anxiety after COVID-19 Infection between Recovered Patients with and without a Psychiatric History: A Cross-Sectional Study

**DOI:** 10.3390/ijerph191811316

**Published:** 2022-09-08

**Authors:** Megumi Hazumi, Kentaro Usuda, Emi Okazaki, Mayumi Kataoka, Daisuke Nishi

**Affiliations:** 1Department of Public Mental Health Research, National Institute of Mental Health, National Center of Neurology and Psychiatry, Tokyo 187-8551, Japan; 2Department of Sleep-Wake Disorder, National Institute of Mental Health, National Center of Neurology and Psychiatry, Tokyo 187-8551, Japan; 3Department of Mental Health, Graduate School of Medicine, The University of Tokyo, Tokyo 113-8654, Japan

**Keywords:** COVID-19, depression, anxiety, post-acute COVID-19 syndrome, post-COVID

## Abstract

Background: This study aimed to examine the course of depression and anxiety in COVID-19 survivors with a psychiatric history compared with those without a psychiatric history. Methods: A web-based cross-sectional survey for COVID-19 survivors was conducted from July to September 2021. A total of 6016 COVID-19 survivors, the accuracy of whose responses was determined to be assured, were included in analyses. Exposures included psychiatric history and time since COVID-19 infection, and the main outcomes and measures included severity of depression and anxiety, as assessed using the Patient Health Questionnaire-9 (PHQ-9) and Generalized Anxiety Disorder-7 (GAD-7), respectively. Results: Mean severity of PHQ-9 and GAD-7 were significantly higher in participants with a psychiatric history than in those without a psychiatric history. Two-way analysis of covariance for PHQ-9 showed a significant main effect of the presence of psychiatric history and a significant interaction effect of psychiatric history × time since infection. Two-way analysis of covariance for the GAD-7 score revealed a significant main effect of the presence of psychiatric history and time since COVID-19 infection and the interaction effect of these factors. Conclusions: The course of depression and anxiety was more severe in COVID-19 survivors with a psychiatric history than in those without a psychiatric history.

## 1. Introduction

Psychiatric symptoms such as depression and anxiety after COVID-19 infection are among the most serious issues that persist during acute, long-COVID, and post-COVID periods [1]. Among COVID-recovered patients, 20.6% had depression and 31.4% had anxiety at 4 months after infection [2]. A meta-analysis also found that, among COVID-recovered patients, 14% to 44% had general anxiety disorder and 19.2% to 21.5% had depression 4 months after infection [3]. Depression and anxiety persisted in 23% of survivors at 6 months and 26% at 12 months after infection, and the proportion of individuals having these symptoms was higher compared with controls [4,5,6]. Furthermore, the risk of mental health problems as sequelae were higher in COVID-recovered patients than in influenza survivors and those with other respiratory infectious diseases [5,6,7]. Depression and anxiety reduce quality of life and can cause tremendous economic loss to society as a whole [8,9,10]. Therefore, it is important to clarify the course of psychiatric symptoms such as depression and anxiety in COVID-recovered patients.

A systematic review of this course pattern revealed conflicting results in the literature [11]. For example, one study reported that the severity of depression in COVID-recovered patients had normalized at 1-month follow-up [12], whereas other studies reported a poor course or no change [7,13,14,15]. As for anxiety, similarly, there are reports of improvement, worsening, and no change, with no agreement on the course [7,12,13,14,15,16]. Given these differences in the course of psychiatric symptoms among studies, the course might be influenced by potential background factors.

Several factors have been found to be associated with psychiatric symptoms after COVID-19, with psychiatric history presumably one of the most important factors. People who have psychiatric histories are more likely to have psychiatric symptoms after an adverse experience [17]. Experiences of COVID-19 infection are extremely painful [18,19,20,21,22], and these distressing experiences affect mental health in COVID-recovered patients indeed [23]. Considering them, COVID-recovered patients with a psychiatric history might be more affected by the adversity of infection and more likely to have current psychiatric symptoms after infection. Furthermore, inflammation with COVID-19 infection is related to depression and anxiety in COVID-recovered patients [24,25,26,27]. Given the neuroinflammation hypothesis of psychiatric disorders, a psychiatric history promotes neuroinflammation in COVID-19 infection. As a result, those with a psychiatric history tend to have more severe depression and anxiety as sequelae of COVID-19 infection by neuroinflammation. Based on these psychological and biological pathways, psychiatric history is presumed to contribute to the course of psychiatric symptoms as COVID-19 sequelae, although it remains unclear whether the course of psychiatric symptoms as sequelae of COVID-19 differs between COVID-recovered patients with a psychiatric history and those without. A study reports that several psychiatric symptoms at 1 and 3 months are already known to be worse in COVID-recovered patients with a psychiatric history than in those without a psychiatric history [12]. However, the course should be confirmed over a longer term, given the reports of psychiatric symptoms lingering after 6 or even 12 months [4], and interaction effects should be examined with a larger sample. If their depression and anxiety are not improved over time as well as severity, they need to be treated rather than waiting for spontaneous remission. Therefore, clarifying the course of their psychiatric symptoms is an important for treatment decision making.

The purpose of this study was to examine whether the severity of psychiatric symptoms over time after COVID-19 infection differs depending on the presence or absence of a psychiatric history before COVID-19 infection. We hypothesized that psychiatric symptoms would improve over time in those without a psychiatric history but worsen over time in those with a psychiatric history. This hypothesis will help determine whether spontaneous remission can be expected or treatment would be necessary for psychiatric symptoms as sequelae of COVID-19.

## 2. Materials and Methods

### 2.1. Participants and Setting

A web-based cross-sectional survey of participants who had been infected with COVID-19 infection was conducted from July 2021 to September 2021 in Japan. Participants were recruited from a major internet survey agency with a survey panel of approximately 2.2 million registered individuals in 2019 (Rakuten Insight, Tokyo, Japan). To advertise the survey panel of the internet survey agency to participate the survey, an email was sent to the survey panel encouraging them to participate in the survey via a survey link posted on the homepage accessible only to the survey panel. The survey link can only be accessed once per person by those registered with the survey panel.

The survey panel was asked to complete the screening questionnaire to identify subjects who meet the inclusion criteria. Those who agreed to respond to the screening questionnaire and met the inclusion criteria were asked to complete the survey questionnaire. Those who responded to the survey questionnaire but met the following exclusion criteria were eluded from the analysis.

We included individuals who responded “yes” to the question “Have you ever been infected with COVID-19?”. Among participants who responded “yes” to the question and completed the questionnaire, we excluded the following from the analyses: (a) those who selected incorrect alternatives to the dummy question (i.e., the fraudulent response in this study); (b) those who disclosed having not been infected while answering the survey; (c) those whose time since infection was outside the range of 0 to 20 months based on the date of introduction of COVID-19 to Japan; (d) those who selected “others” for the question about educational attainment and did not reclassify their educational attainment as a free-text response. In addition, those who discontinued their responses to the questionnaire were excluded from the analysis.

### 2.2. Ethics Statements

This study was approved by the ethical board of the National Center of Neurology and Psychiatry in Japan (A2021-34). Informed consent was obtained electronically from all participants prior to the survey. The study conformed to the Strengthening the Reporting of Observational Studies in Epidemiology (STROBE) guidelines [28].

### 2.3. Measurement

#### 2.3.1. Outcome Variables

Two scales were used to measure depression and anxiety. Severity of depression was measured by using the Patient Health Questionnaire-9 (PHQ-9), which has high validity and reliability [29,30] and consists of 9 items rated on a 4-point scale: (0) not at all, (1) several days, (2) more than half the days, and (3) nearly every day.

The severity of anxiety was measured using the Generalized Anxiety Disorder-7 (GAD-7), which has high validity and reliability [31,32] and consists of 7 items rated on the same 4-point scale as the PHQ-9. In both scales, a higher total score indicates greater symptom severity.

#### 2.3.2. Exposure Variables

Exposure variables were psychiatric history and time since COVID-19 infection.

The presence of psychiatric history was classified according to answers to the question “Have you ever been diagnosed with or experienced psychiatric problems before the COVID-19 pandemic?”. For this question, multiple answers were allowed from the following: “Nothing”, “Depressive disorder”, “Bipolar disorder”, “Panic attack or panic disorder”, “Anxiety disorder or anxiety-related problems (e.g., hypersensitivity, worry, fear, obsessive-compulsive symptoms)”, “Alcohol use disorder or alcohol abuse/dependence”, “The use of illicit substances or psychotropics without prescription”, “Burnout syndrome”, and “Others” (with a column for write-in details). Based on the Diagnostic and Statistical Manual of Mental Disorders Fifth Edition (DSM-5) [33], these options were categorized into those with and without psychiatric history. Specifically, those who chose options other than “Nothing”, “The use of illicit substances or psychotropics without prescription”, and “Burnout syndrome” were classified into the group of those with psychiatric history. “The use of illicit substances or psychotropics without prescription” was not considered to meet the diagnostic criteria for alcohol use disorder by itself (long-term use, heavy use, craving, etc.), and those who selected only this option were classified as the group of those without a psychiatric history. Furthermore, since the mental condition of “Burnout syndrome” is not included in the DSM-5, participants who selected only this option were classified as the group of those without a psychiatric history. Among those who selected “Others” alone, those whose comments in the columns matched DSM-5 diagnosis were classified into the group of those with psychiatric history, and those who did not match were classified into the group of those without a psychiatric history.

Time since COVID-19 infection was defined as the period from the date of discovery of infection to the date of response to the questionnaire and was classified according to 4 periods (<1 month, ≥1 month, ≥3 months, ≥6 months). This classification was based on the definition of symptoms that occurred at less than 1 month as “acute”, symptoms between 1 and 3 months as “long COVID”, and symptoms over 3 months as “post-COVID” [1,34].

#### 2.3.3. Covariates

We included the following variables, which are suggested to be associated with exposure variables and outcome variables as covariates: income level based on low income and median income definitions by the Ministry of Health, Labor and Welfare (low, <3 million JPY; median, <10 million JPY; high, ≥10 million JPY; unknown or refused to answer) [35,36,37], residence (alone, co-residence) [38,39], and comorbidities (hypertension, diabetes, asthma, angina pectoris or cardiac infarction, chronic obstructive pulmonary disease, chronic pain) [40,41]. Additionally, the presence of physical sequelae symptoms was used as a covariate for a sensitivity analysis.

Demographic information was also collected and included as covariates such as age group (20–29, 30–39, 40–49, 50–59, 60–69, 70–79, 80–89, ≥90), sex (male, female, other), educational attainment (high school education or lower, higher than high school).

These covariates were converted into a binary dummy variable except for age groups, and considered acceptable for parametric analysis because the residual error was almost normally distributed.

### 2.4. Statistical Analysis

Mean and standard deviation (SD) were calculated for continuous variables, and count and percentage were calculated for categorical variables. The χ^2^-test for categorical variables and *t*-test for continuous variables were used for comparisons between participants without a psychiatric history and those with a psychiatric history.

Mean and standard error were calculated for each group of psychiatric history and time since COVID-19 infection. Two-way analysis of variance (ANOVA) and two-way analysis of covariance (ANCOVA) were performed to compare differences in the course of depression and anxiety among participants with and without psychiatric history, regardless of covariates. A simple-effects test was performed using Bonferroni’s method. Sensitivity analyses were performed using two-way ANCOVA with covariates for presence of sequelae symptoms instead of comorbidities. Complete case analysis was used to deal with missing data. The data of those who discontinued their responses were excluded from the analysis.

The significance level was set at *p* < 0.05. All statistical analyses were performed using IBM SPSS statistics version 25 (Chicago, IL, USA).

## 3. Results

### 3.1. Participant Characteristics

Among 9505 individuals who reported being COVID-recovered patients, 6016 were included in the analyses (Figure 1).

Table 1 shows the participant characteristics in this study. In total, mean PHQ-9 and GAD-7 scores were 5.19 (SD = 5.91) and 3.34 (SD = 4.71), respectively. The severity of depression and anxiety was significantly higher in participants with a psychiatric history than in those without a psychiatric history (PHQ-9, 9.74 (SD = 7.02) vs. 4.21 (SD = 5.14), *p* < 0.001; GAD-7, 7.09 (SD = 5.85) vs. 2.54 (SD = 4.0), *p* < 0.001). Mean time since COVID-19 infection was 4.5 (SD = 4.37) months, which was significantly longer in participants with a psychiatric history than in those without a psychiatric history (mean 4.44 (SD = 4.36) vs. mean 4.79 (SD = 4.42), *p* = 0.018).

### 3.2. Depression at Each Period in COVID-Recovered Patients with and without a Psychiatric History

In participants with a psychiatric history, the PHQ-9 mean score was 8.95 (SE = 0.40) at <1 month, 8.42 (SE = 0.36) at ≥1 month, 9.69 (SE = 0.33) at ≥3 months, and 9.23 (0.27) at ≥6 months. In participants without a psychiatric history, the mean score was 5.02 (SE = 0.17) at <1 month, 4.44 (SE = 0.15) at ≥1 month, 4.01 (SE = 0.16) at ≥3 months, and 4.10 (SE = 0.13) at ≥6 months.

Two-way ANOVA for the PHQ-9 revealed a significant main effect of psychiatric history and significant interaction effect of psychiatric history × time since COVID-19 infection (SS = 23,979.86, df = 1, MS = 23,979.86, F = 791.81, η^2^ = 0.12, *p* < 0.001; SS = 541.56, df = 3, MS = 180.52, F = 5.96, η^2^ = 0.003, *p* < 0.001). The main effect of psychiatric history and the interaction effect of psychiatric history × time since COVID-19 infection were maintained at ANCOVA (Table 2). These main and interaction effects were maintained after adjusting for additional covariates such as the presence of sequelae (Appendix A).

A simple-effects test revealed a significant difference between participants with a psychiatric history and those without a psychiatric history in all periods (*p* < 0.001). As Figure 2 shows, in participants without a psychiatric history, the severity of depression was significantly higher at <1 month than at ≥3 months and ≥6 months (both *p* < 0.001). In participants with a psychiatric history, significant differences were found between the periods.

### 3.3. Anxiety at Each Period in COVID-Recovered Patients with and without a Psychiatric History

In participants with a psychiatric history, the mean GAD-7 score was 5.80 (SE = 0.32) at <1 month, 5.78 (SE = 0.28) at ≥1 month, 7.27 (SE = 0.27) at ≥ 3 months, and 7.02 (SE = 0.12) at ≥6 months. In participants without a psychiatric history, the mean score was 2.76 (SE = 0.14) at <1 month, 2.65 (SE = 0.12) at ≥1 month, 2.61 (SE = 0.13) at ≥3 months, and 2.56 (SE = 0.11) at ≥6 months.

Two-way ANOVA for GAD-7 revealed a significant main effect of psychiatric history and time since COVID-19 infection, as well as an interaction effect of these factors (SS = 15,712.37, df = 1, MS = 15,712.37, F = 824.14, η^2^ = 0.12, *p* < 0.001; SS = 284.56, df = 3, MS = 94.85, F = 4.98, η^2^ = 0.002, *p* = 0.002; SS = 478.57, df = 3, MS = 159.52, F = 8.37, η^2^ = 0.004, *p* < 0.001). These relationships remained after adjusting for covariates at ANCOVA (Table 2). These main and interaction effects were maintained after adjusting for additional covariates such as the presence of sequelae (Appendix A).

As Figure 2 shows, a simple-effects test for GAD-7 showed a significant difference between participants with a psychiatric history and participants without a psychiatric history at all times since COVID-19 infection (*p* < 0.001). Anxiety in participants with a psychiatric history was lower at <1 month than at ≥3 months and ≥6 months (≥3 months, *p* < 0.001; ≥6 months, *p* = 0.002), and lower at ≥1 month than at ≥3 months and ≥6 months (≥3 months, *p* = 0.001; ≥6 months, *p* = 0.005). No significant differences were found in participants without a psychiatric history.

## 4. Discussion

To our knowledge, this is the second study to examine differences in the course of depression and anxiety after COVID-19 infection between those with and without a psychiatric history. In this study, the sample size was much larger and the follow-up period was much longer than in the first study [12]. Our cross-sectional survey indicates that the severity of depression persists over time in COVID-recovered patients with a psychiatric history, whereas it decreases gradually with time in those without a psychiatric history. In contrast, the severity of anxiety increased progressively in COVID-recovered patients with a psychiatric history but not in those without a psychiatric history.

The course of depression and anxiety was indicated to be favorable over time in COVID-recovered patients without a psychiatric history, but not in those with a psychiatric history. This finding was consist with the studies by Taquet et al. and Matalon et al., which suggested that the risk of psychiatric symptoms such as depression and anxiety in COVID-recovered patients decreases after 1 month [7,12]. In the present study, depressive symptoms remained unchanged and anxiety symptoms worsened in COVID-recovered patients with a psychiatric history, which is similar to the results of Iqbal et al. and Lorenzo et al., suggesting that psychological symptoms do not improve with time [14,15]. A systematic review of the course of depression in COVID-recovered patients found conflicting results [11], possibly due to different proportions of participants with a psychiatric history included in each study. A previous study failed to find an interaction effect of psychiatric history and time since COVID-19 infection, with depression maintained regardless of psychiatric history [12]. In that study, depression within the first 3 months was similar to that in Mazza’s study [12]. Given this similarity, the present study explored a longer course than the previous one, revealing the difference in the course of depression severity between those with and without a psychiatric history. Therefore, early detection and care might require monitoring depression and anxiety of those with a psychiatric history for a more extended period (e.g., 6 months or more). The bidirectional relationship between COVID-19 symptoms and psychiatric disorders have been clarified [5]: while COVID-19 infection predisposes to psychiatric disorder [2,3,4,5], having psychiatric disorders is a risk factor of post-acute sequelae of COVID-19 infection [17]. Our findings are thought to add the perspective of change over time to this relationship.

Our finding might be explained from the view of psychosocial and biological perspective. In the view of psychosocial factors, COVID-recovered patients can often experience various psychosocial burdens such as stigma [18,19], feeling guilty [20,21], and concern about symptom reactivation and reinfection [19]. These psychosocial stressors affect psychiatric symptoms in COVID-recovered patients [22]. While higher resilience contributes lowers severity of psychiatric symptom in COVID-recovered patients [42]. Considering individuals with a psychiatric history tend to be less resilient in general [17], COVID-recovered patients with a psychiatric history might need more time to recover their mental health due to reduced resilience in managing an adverse experience such as a COVID-19 infection [27].

Biological factors are another possible factor. For example, residual inflammation, and microvascular dysregulation, the gut microbiome and autoimmune phenomena are recently found to lead to COVID-19 sequelae including neuropsychiatric symptoms [27]. In particular, some studies indicate that increased inflammation associated with COVID-19 is related with subsequent depressive and anxiety symptoms [25,26,27]. It is widely known that inflammation is one of the etiological factors of psychiatric disorders. Therefore, people with a psychiatric history might be more likely to experience worsening or prolonged psychiatric symptoms such as depression and anxiety due to inflammation. Given the neurodegeneration associated with the brain’s inflammatory response [24,25,26], as well as the biological mechanisms involved, an individual with a psychiatric history is unlikely to recover in a short period of time.

This study has some limitations. First, the presence or absence of infection was anonymously confirmed by self-report. Furthermore, self-reported diagnoses sometimes differ from those made by physicians and may be incorrect due to misunderstanding or memory impairment. However, the survey contractor took measures to prevent false negatives through incentives for respondents as well as a sufficient sample size. Additionally, the incidence rate of infected cases among all screening respondents in this survey was 1.93%, close to the cumulative rate of infected cases at that time, which was about 1.4% [43]. Therefore, we can assume that there was no remarkable discrepancy. Nevertheless, the factuality and objectivity of the data are still limited, given the self-reporting and recall bias. Second, this was a cross-sectional survey, and thus causal inferences cannot be made. In particular, it is necessary to assume the impact of differences in social environment at different times of infection, such as phases of the infection spread situation or amount of accurate information about COVID-19. However, a cross-sectional self-report survey was the only way to quickly and clearly show the progress of each period of time since infection with such a large sample size. Third, sampling bias should be mentioned. Participants were recruited from a specific survey panel of an internet survey agency, and participants’ condition was not so severe such that they had enough vitality to respond to the enormous questionnaires. Furthermore, considering over 90% of participants were under 60 years old, results in this study are not appropriate to apply to the elderly population. Fourth, remission status or pre-infection symptom severity in those with psychiatric history was unclear in this study. There is some possibility that depression and anxiety in those with psychiatric history observed in this study represent symptoms that continued from before infection rather than newly emerging after infection. Fifth, confounding factors were not fully adjusted (e.g., living in an urban or rural area). Thus, unmeasured confounding factors may potentially affect the results in this study. In addition, we did not consider age itself in this study for ethical consideration and had to use the age group as a covariate. However, the distribution of this variable may deviate somewhat from a normal distribution. Finally, the degree of improvement in PHQ-9 in those without a psychiatric history may be too small to interpret as clinically significant change [44].

## 5. Conclusions

Our study revealed that the severity of depression and anxiety as COVID-19 sequelae might progressively decrease or remain low in COVID-recovered patients without a psychiatric history, whereas these might remain constant or worsen in COVID-recovered patients with a psychiatric history. In COVID-recovered patients with a psychiatric history, subsequent depression and anxiety might not remit spontaneously. Therefore, COVID-recovered patients with psychiatric histories should be carefully monitored and provided special care. Since this study has several limitations mentioned above, longitudinal studies should be conducted in the future to further verify the implications found in this study.

## Figures and Tables

**Figure 1 ijerph-19-11316-f001:**
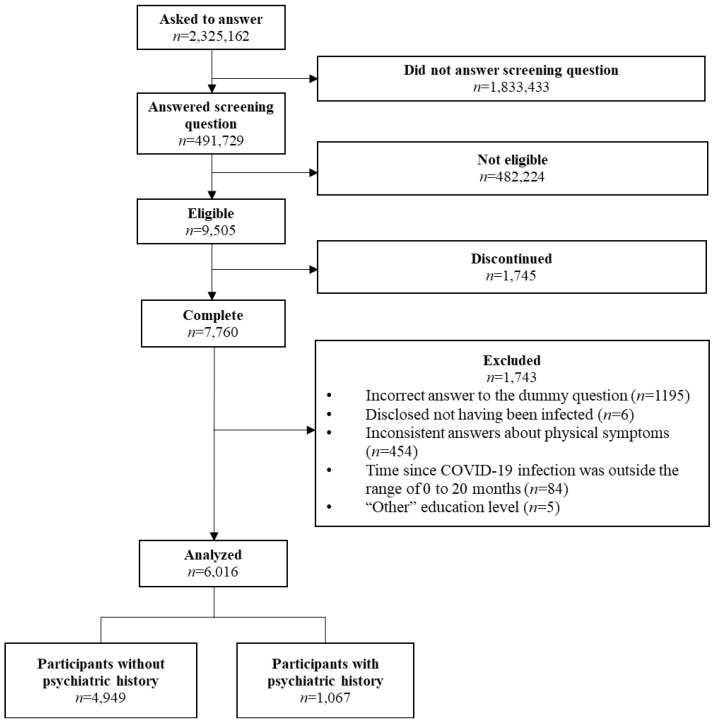
Flow chart of the participants from recruitment to analysis.

**Figure 2 ijerph-19-11316-f002:**
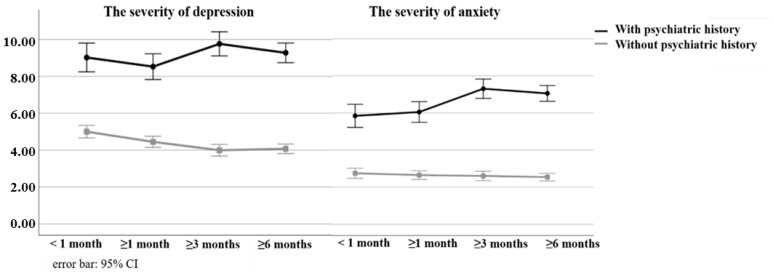
Course of the severity of depression and anxiety after infection.

**Table 1 ijerph-19-11316-t001:** Participant characteristics.

	Total		Participants without Psychiatric History		Participants with Psychiatric History			
	*n* = 6016		*n* = 4949		*n* = 1067			*p*
**Psychiatric history**	1067	17.73%						
Depressive disorder					600	56.23%		
Bipolar disorder					211	19.78%		
Panic attack or panic disorder					228	21.37%		
Anxiety disorder or anxiety-related problems					351	32.90%		
Alcohol use disorder or alcohol abuse/dependence					92	8.62%		
Others					47	4.40%		
**Time since COVID-19 infection (month)**	4.50	±4.37	4.44	±4.36	4.79	±4.42	t = −2.38	*p* = 0.02
<1 month	1138	18.92%	956	19.32%	182	17.06%	χ^2^ = 11.10	*p* = 0.01
<3 months	1443	23.99%	1216	24.57%	227	21.27%		
<6 months	1369	22.76%	1108	22.38%	261	24.46%		
≥6 months	2066	34.34%	1669	33.72%	397	37.21%		
**PHQ-9**	5.19	±5.91	4.21	±5.14	9.74	±7.02	t = −24.38	*p* < 0.001
≥10	1196	19.88%	695	14.04%	501	46.95%	χ^2^ = 596.88	*p* < 0.001
**GAD-7**	3.34	±4.71	2.54	±4	7.09	±5.85	t = −24.24	*p* < 0.001
≥10	690	11.47%	352	7.11%	338	31.68%	χ^2^ = 521.65	*p* < 0.001
**Age group (years)**								
20–29	1192	19.81%	987	19.94%	205	19.21%	χ^2^ = 40.01	*p* < 0.001
30–39	1505	25.02%	1210	24.45%	295	27.65%		
40–49	1581	26.28%	1260	25.46%	321	30.08%		
50–59	1189	19.76%	994	20.08%	195	18.28%		
60–69	436	7.25%	395	7.98%	41	3.84%		
70–79	101	1.68%	91	1.84%	10	0.94%		
80–89	7	0.12%	7	0.14%	0	0.00%		
≥90	5	0.08%	5	0.10%	0	0.00%		
**Sex**								
Male	3441	57.20%	2832	57.22%	610	57.17%	χ^2^ = 0.07	*p* = 0.97
Female	2555	42.47%	2101	42.45%	453	42.46%		
Others	20	0.33%	16	0.32%	4	0.37%		
**Educational attainment**								
High school educated or lower	1355	22.52%	1087	21.96%	268	25.12%	χ^2^ = 5.00	*p* = 0.03
Higher than high school educated	4661	77.48%	3862	78.04%	799	74.88%		
**Income level**								
Low	842	14.00%	638	12.89%	204	19.11%	χ^2^ = 31.47	*p* < 0.001
Medium	3626	60.27%	3000	60.61%	626	58.67%		
High	988	16.42%	841	16.99%	147	13.78%		
Unknown or refused to answer	560	9.31%	470	9.50%	90	8.43%		
**Residence**								
Alone	1165	19.37%	947	19.14%	218	20.43%	χ^2^ = 0.94	*p* = 0.33
Co-residence	4851	80.63%	4002	80.86%	849	79.57%		
**Comorbidities**								
Hypertension	892	14.83%	688	13.90%	204	19.12%	χ^2^ = 18.92	*p* < 0.001
Diabetes	378	6.28%	263	5.31%	115	10.78%	χ^2^ = 44.50	*p* < 0.001
Asthma	277	4.60%	166	3.35%	111	10.40%	χ^2^ = 99.29	*p* < 0.001
Angina pectoris or cardiac infarction	135	2.24%	73	1.48%	62	5.81%	χ^2^ = 75.22	*p* < 0.001
Chronic obstructive pulmonary disease	74	1.23%	35	0.66%	39	3.71%	χ^2^ = 62.78	*p* < 0.001
Chronic pain	940	15.63%	667	13.48%	273	25.59%	χ^2^ = 97.70	*p* < 0.001

**Table 2 ijerph-19-11316-t002:** Results of ANCOVA for Depression and Anxiety.

	Depression (PHQ-9)						Anxiety (GAD-7)					
	*SS*	*df*	*MS*	*F*	*η^2^*	*p*	*SS*	*df*	*MS*	*F*	*η^2^*	*p*
Psychiatric history	17,773.61	1.00	17,773.61	623.55	0.094	<0.001	12,068.84	1.00	12,068.84	662.65	0.10	<0.001
Time since COVID-19 infection	115.27	3.00	38.42	1.35	0.001	0.26	249.10	3.00	83.03	4.56	0.002	0.003
Psychiatric history × Time since COVID-19 infection	432.98	3.00	144.33	5.06	0.003	0.002	375.75	3.00	125.25	6.88	0.003	<0.001

PHQ-9, Patient Health Questionnaire-9; GAD-7, General Anxiety Disorder-7.

## Data Availability

The data used in this study will be made available to the corresponding author upon reasonable request.

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
