# Peer review of "Differences in the Course of Depression and Anxiety after COVID-19 Infection between Recovered Patients with and without a Psychiatric History: A Cross-Sectional Study"

_ijerph, 2022, doi:10.3390/ijerph191811316_

Round 1

Reviewer 1 Report

The article is very interesting, top quality and significant for improving our knowledge.

I highly recommend it for the Journal.

The main aim of the research was to find difference in post-covid health problems between people suffering from mental disorders before getting sick

The topic is relevant and very intesing due to the method used. In my perspective the use of population data from the Japanese database is of great value and allows for the generalization of results.

The topic is not very original due to the fact that you can observe a common trend for covid research in  many field, but on the other hand, this trend is a result of the need to fill in gaps in the existing knowledge. So the topic itself is not original, but thanks to the methodology used, the research is in my opinion very original.

In my opinion, the limitation of the study is the fact that clinical diagnosis of mental disorders was not used as a criterion for differentiating between groups, but only a subjective answer to criteria questions. And this may be a source of errors, as the memory of the respondents may be unreliable as to their previous disease experiences - especially since postovid disorders also relate to memory problems. However, due to the size of the sample, it seems that such error can be ignored

The study extend our knowledge based on the Matalon et al (2021) reaseach due to the sample size was much larger and the follow-up period was much longer than only the one month as in the mentioned study.

In my opinion conclussions are consisten with results and arguments discussed. The study confirmed on a large sample that symptoms of depression and anxiety as a consequence of COVID-19 are more evident in patients reporting mental health disorders before becoming ill on covid. The study gives us stotrong practical conclusion, wha is very important that such patients should be given special care and monitoring.

is about the detaild on the mental health problems (p. 3). “Have you

ever been diagnosed with or experienced psychiatric problems before the COVID-19 pan demic?” The is burnout syndrome mentioned as the example. If I understand properly it was the example to support the participants to interprete their mental problems however it is not the deasise. It would be explained more effectively. 

To be fair with you I am not qualified with bid data statistics and I am not competent to authoritatively state whether the applied statistics are adequate to the data. The first reviewer's critical remarks relate to this issue and gave strong aguments. However, I would like to emphasize with full conviction that the text is convincing and reliably prepared.

Best regards,

Reviewer 2 Report

I believe it is a valuable work to investigate differences in the course of depression and anxiety after COVID-19 infection between patients with psychiatric history and those who do not. I think it will be necessary to revise and supplement it to be published in the Q1 or Q2 journal. Those things needed to be revised are as follows.

1. Introduction: It is necessary to explain in more detail the rationale that raised the hypothesis or research problem of this study. For example, having psychiatric history means that symptoms are severe or that there are no other resources to help overcome those symptoms, so symptoms are worse or hard to be recovered from, and you should explain the need to conduct this study.

2. You adjusted the ANOVA models with several covariates, and you should also suggests rationales. If they do not satisfied conditions of the exogenous confounding variable, the model is distorted because of Berkson’s paradox. The condition of the exogenous confounding variable must be related to both the independent variable (predictor) and the dependent variable (criterion variable).

3. And, you didn't include how the variables defined as covariates were put in dummy variables in parametric statistical analyses, because they are non-parametric. Age is a parametric data, but you should not forget that age groups are even non-parametric variables.

4. One of the limitation of this study is that there are too many samples to conclude with inferential statistical analyses. So, you may not have given the mean scores, but do you think the statistically significant difference in mean scores of depression and anxiety between groups is clinically significant?

 Anyway, I expected that you can revise the manuscript according to I mentioned above.

Round 2

Reviewer 2 Report

Thank you for revising the manuscript well regarding my comment.
I think it would be good to supplement it carefully so that the quality of the manuscript can be improved before publication if it wiil be accepted.

Author Response

Thank you for revising the manuscript well regarding my comment.

I think it would be good to supplement it carefully so that the quality of the manuscript can be improved before publication if it will be accepted.

Reply:

Thank you for your comment. Based on your evaluation that “Are the conclusions supported by the results?→Can be improved” at the reviewing checklist, we revised conclusion as follows with attention to clarity and understandability. Modified parts were highlighted with yellow markers.

 In case this revision is not sufficient, we would appreciate it if you kindly tell us specifically what revisions are needed. We would like to reflect your point in the final version.

(Page 9, Line 32 to 40, Conclusions)

Conclusions

Our study revealed that the severity of depression and anxiety as COVID-19 sequelae might progressively decrease or remain low in COVID-recovered patients without a psychiatric history, whereas these might remain constant or worsen in COVID-recovered patients with a psychiatric history. In COVID-recovered patients with a psychiatric history, subsequent depression and anxiety might not remit spontaneously. Therefore, COVID-recovered patients with psychiatric histories should be carefully monitored and provided special care. Since this study has several limitations mentioned above, longitudinal studies should be conducted in the future to further ensure the implications found in this study.